# A Robust Backpropagation-Free Framework for Images

**Timothy Zee**  *tsz2759@rit.edu*
*Department of Computer Science*
*Rochester Institute of Technology*

**Alexander G. Ororbia**  *ago@cs.rit.edu*
*Department of Computer Science, Department of Psychology*
*Rochester Institute of Technology*

**Ankur Mali**  *ankurarjunmali@usf.edu*
*Department of Computer Science and Engineering*
*University of South Florida*

**Ifeoma Nwogu**  *inwogu@buffalo.edu*
*Department of Computer Science*
*University at Buffalo, SUNY*

**Reviewed on OpenReview:** *https://openreview.net/forum?id=leqrOvQzeN*

## Abstract

While current deep learning algorithms have been successful for a wide variety of artificial intelligence (AI) tasks, including those involving structured image data, they present deep neurophysiological conceptual issues due to their reliance on the gradients that are computed by backpropagation of errors (backprop). Gradients are required to obtain synaptic weight adjustments but require knowledge of feed-forward activities in order to conduct backward propagation, a biologically implausible process. This is known as the "weight transport problem". Therefore, in this work, we present a more biologically plausible approach towards solving the weight transport problem for image data. This approach, which we name the error-kernel driven activation alignment (EKDAA) algorithm, accomplishes through the introduction of locally derived error transmission kernels and error maps. Like standard deep learning networks, EKDAA performs the standard forward process via weights and activation functions; however, its backward error computation involves adaptive error kernels that propagate local error signals through the network. The efficacy of EKDAA is demonstrated by performing visual-recognition tasks on the Fashion MNIST, CIFAR-10 and SVHN benchmarks, along with demonstrating its ability to extract visual features from natural color images. Furthermore, in order to demonstrate its non-reliance on gradient computations, results are presented for an EKDAA-trained CNN that employs a non-differentiable activation function. Our library implementation can be found at: *https://github.com/tzee/EKDAA-Release*.

# 1 Introduction

One of the most daunting challenges still facing neuroscientists is the understanding of how the neurons in the complex network that underlies the brain work together and adjust their synapses in order to accomplish goals (Südhof & Malenka, 2008). While artificial neural networks (ANNs) trained by backpropagation of errors (backprop) present a practical, feasible implementation of learning by synaptic adjustment, it is largely regarded by neuroscientists as biologically implausible for various reasons, including the implausibility of the direct backwards propagation of error derivatives for synaptic updates. Furthermore, this form of learning breaks a fundamental requirement for biologically-plausible (bio-plausible) learning; backprop requires access to the feed-forward weights in order pass back error signals to previous layers. This property, known as the *weight transport problem* (Grossberg, 1987), renders backprop to be a poor candidate base learning rule for building realistic bio-inspired neural modeling frameworks. In terms of bio-plausibility, it is more likely that differences in neural activity, driven by feedback connections, are used in locally effecting synaptic changes (Lillicrap et al., 2020). Such difference-based networks overcome some of backprop's major implausibilities in a way that is more naturalistic and more compatible with the current understanding of how brain circuitry operates. Although a few classes of algorithms have been proposed to address the specific challenge of error gradient propagation in training ANNs, fewer still have been proposed to handle the highly structured data found in large-scale image datasets. Current-day convolutional neural networks (CNNs) and Visual Transformeres (ViTs) continue to set the benchmark standards for difficult vision problems (Mnih et al., 2013; He et al., 2016; Dosovitskiy et al., 2020), and they do so using backprop, with symmetric weight matrices in both the feedforward and feedback pathways.

## 1.1 Bio-plausible Machine Learning

In this work, we create a learning rule that addresses the weight transport problem for image data, leveraging spatial relationships with convolution. As a result, we introduce a more bio-plausible error synaptic feedback mechanism that we deem the (learnable) *error-kernel*, which generates target activities for feature maps within a CNN to "align" to. We call this learning mechanism *error-kernel driven activation alignment (EKDAA)*.

**The Weight Transport Problem.** In our learning scheme, the forward pathway relies on traditional weight matrices/tensors whereas the backward pathway focuses on error kernels and maps, thus eliminating two-way symmetric weight structure inherent to backprop-trained networks. This, in effect, resolves the weight transport problem.

While currently known bio-plausible methodologies have not reached the modeling performance of backprop and have yet to be scaled to large datasets such as ImageNet (Deng et al., 2009), we believe investigating bio-plausible learning rules are key in the future of neural modeling. EKDAA notably opens the door to a wider variety of neural structures, potentially enabling lateral neural connections, where forward/backward propagation no longer carry the traditional meaning. We successfully train a convolutional network on image data, using the signum function (which has a derivative of zero everywhere except at zero, i.e., its derivative is a Dirac delta function). Learning with the signum showcases how EKDAA facilitates the use of bio-plausible activation functions. The signum function behaves similarly to the action potential of a biophysical neuron, i.e., it acts as a hard activation where the incoming signal is either propagated or killed, abiding to Dale's law for neurons (Eccles, 1976; Lillicrap et al., 2020).

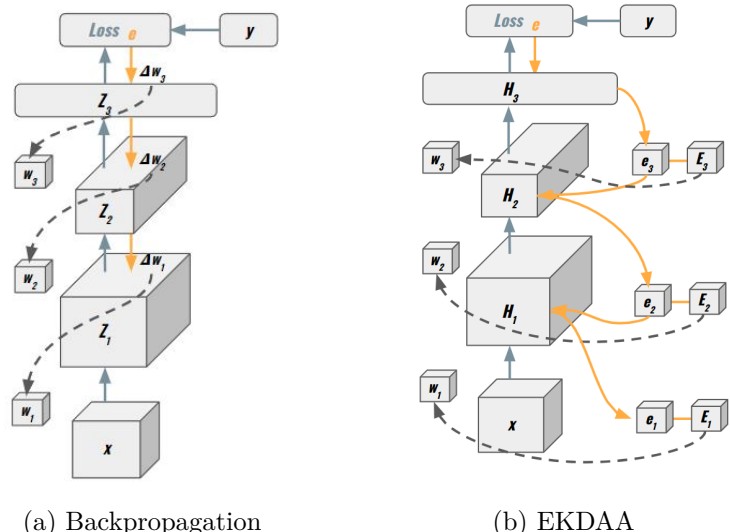

(a) Backpropagation        (b) EKDAA

Figure 1: The signal flow for backprop and EKDAA. The forward pass (solid blue arrows), the backward pass (solid orange arrows), and the weight update (gray dashed lines). For layer $N$, $H_N$ denotes its pre-activation while $Z_N$ represents its post-activation. $e$ is the convolutional error kernel, and $E$ is used to transpose the error signal to the appropriate size when propagating backwards.

**Bio-plausibility and Convolution.** The convolution operator is a powerful mechanism for extracting spatial features from images and video, exhibiting key properties for signal updates that can be integrated into a bio-plausible convolutional network. In a backprop model, deconvolution is used for both layer-wise gradient updates and deconvolution of filter updates, and can be computed by differentiating each value of the filter with respect to every element of the matrix that the filter touches during the convolution pass. However, gradient updates can also be computed without the need to take direct derivatives on elements by directly employing a convolution of the same matrices that the derivatives would be taken on. As a result, the update signals are computed as:

$$\mathbf{\Delta W}^{\ell}_{\mathbf{m,n,:,:}} \leftarrow \mathbf{X}^{\ell}_{\mathbf{m,:,:}} * \mathbf{\Delta X}^{\ell+1}_{\mathbf{m,:,:}} \tag{1}$$

$$\mathbf{\Delta X}^{\ell}_{\mathbf{m,n,:,:}} \leftarrow \mathbf{\Delta X}^{\ell+1}_{\mathbf{m,n,:,:}} * Flip(\mathbf{W}^{\ell}_{\mathbf{m,n,:,:}}) \tag{2}$$

where, $\mathbf{W}^{\ell}_{\mathbf{m,n,:,:}}$ is the weight matrix for a layer $l$ with feature map element $[m, n]$, $\mathbf{X}^{\ell}_{\mathbf{m,:,:}}$ is the corresponding model layer output. $\mathbf{\Delta W}^{\ell}_{\mathbf{m,n,:,:}}$ is the error signal weight matrix for layer $l$, $\mathbf{\Delta X}^{\ell}_{\mathbf{m,:,:}}$ is the error signal from the output of layer $l$, and $Flip$ is the transpose of a matrix with respect to both the $x$ and $y$ axis. The convolution operator is represented with $*$. Therefore, convolution works within a Hebbian model and can be used as a powerful feature extractor without the need for computed derivatives. The symmetrical process of convolution in the inference and training passes may further be considered to be a more bio-plausible means of processing visual information in comparison to the dense transforms used in standard feed-forward models. Convolution and deconvolution also are operators that are agnostic as to whether a learning rule is or is not weight symmetric. In essence, there are no inherent rules that dictate that deconvolution must deconvolve on the exact same weight matrices that are convoluted on in a system's forward pass.

## 2 Related Work

Although Hebbian learning (Hebb, 1949) is one of the earliest and simplest biologically plausible learning rules for addressing the credit assignment problem, extending them to the CNN has not yet been well-developed. Our proposed approach aims to fill this gap.

In addition to Hebbian-based learning, other work includes alternative convolution-based schemes, such as those found in (Akrout et al., 2019), that are based on the Kollen-Pollack (KP) method, which have yielded promising results on larger, more extensive benchmarks without the need for weight transport. These schemes create *weight mirrors* based on the KP method and incorporate it into the convolutional framework. Other training approaches are based on local losses (Nøkland & Eidnes, 2019; Grinberg et al., 2019; Guerguiev et al., 2017; Schmidhuber, 1990; Werbos, 1982; Linnainmaa, 1970); are methods that take the sign of the forward activities (Xiao et al., 2018); are schemes that utilize noise-based feedback modulation (Lansdell et al., 2019); or use synthetic gradients (Jaderberg et al., 2017) to stabilize learning for deeper networks. These approaches have demonstrated better or comparable performance (to backprop) on challenging benchmarks that require the use of convolution. However, there are significant dependencies in the model's forward activities used to guide the backward signal propagation, hence, these approaches belong to a different class of problems/algorithms than what we address in this work.

**The Bottleneck Approach.** Another key effort in this area, inspired by information theory, is the Hilbert-Schmidt independence criterion (HSIC) bottleneck algorithm (Ma et al., 2019), based on the Information Bottleneck (IB) principle (Tishby et al., 2000). HSIC performs credit assignment locally and layer-wise, seeking hidden representations that exhibit high mutuality with target values and less mutuality with inputs (presented) to that layer, i.e., this scheme is not driven by the information propagated from the layer below. Approaches based on the bottleneck mechanism are considered to be the least bio-plausible. Other efforts (Salimans et al., 2017) utilize an evolutionary strategy to search for optimal weights without gradient descent. Although powerful, these approaches exhibit slow convergence, requiring many iterations in order to find optimal solutions.

**Feedback Alignment.** Notably, an algorithm named *Random Feedback Alignment (RFA)* was proposed in (Lillicrap et al., 2016), where it was argued that the use of the transpose of the forward weights ($\mathbf{W}^\ell$ for any layer $\ell$) in backprop, meant to carry backwards derivative information, was not required for learning. This work showed that network weights could be trained by replacing the transposed forward weights with fixed, random matrices of the same shape ($\mathbf{B}^\ell$ for layer $\ell$), ultimately side-stepping the weight transport problem (Grossberg, 1987).

**Direct Feedback Alignment (DFA) .** (Nøkland, 2016), and its variants (Han et al., 2020; Crafton et al., 2019; Chu et al., 2020), was inspired by RFA (Lillicrap et al., 2016), but, in contrast, DFA directly propagates the error signal; it creates a pathway directly between the output layer to internal layers as opposed to a layer-wise wiring format, as in RFA. Notably, across multiple architectures, it was observed that networks trained with DFA showed a steeper reduction in the classification error when compared to those trained with backprop. To compare these bio-plausible feedback alignment-based training paradigms with EKDAA, we extended the corresponding published efforts and implemented CNN versions of FA, DFA, and other related variants. Details of their performance on the several benchmark image datasets investigated in this study are provided in Section 4.

**Target Propagation.** Target propagation (target prop, or TP) (Lee et al., 2015) is another approach to credit assignment in deep neural networks, where the goal is to compute targets that are propagated backwards to each layer of the network. Target prop essentially designs each layer of the network as an auto-encoder, with the decoder portion attempting to learn the inverse of the encoder, modified by a linear correction to account for the imperfectness of the auto-encoders themselves. This corrected difference (between encoder and decoder) is then propagated throughout the network. This process allows difference target prop (DTP) (Lee et al., 2015) and variants (Bartunov et al., 2018; Ororbia & Mali, 2019) (e.g., DTP-$\sigma$) to side-step the vanishing/exploding

gradient problem. However, TP approaches are expensive and can be unstable, requiring multiple forward/backward passes in each layer-wise encoder/decoder in order to produce useful targets.

**Representation Alignment.** *Local Representation Alignment (LRA)* (Ororbia et al., 2018) and recursive LRA (Ororbia et al., 2023) represent yet another class of credit assignment methods, inspired by predictive coding theory (Rao & Ballard, 1999; Clark, 2015; Salvatori et al., 2023) and is similar in spirit to target prop. Under LRA, each layer in the neural network has a target associated with it such that changing the synaptic weights in a particular layer will help move layer-wise activity towards better matching a target activity value. LRA was shown to perform competitively to other local learning rules for fully-connected models, but extending/applying it to vectorized natural images like CIFAR-10 resulted in significant performance degradation.

## 3    Error Kernel Credit Assignment

In implementing the EKDAA algorithm, desirably, the forward pass in the CNN remains the same. However, the backward pass uses a form of Hebbian learning that creates targets for each layer (as shown in Figure 1) which results in an error signal that can be used to make weight adjustments. In convolutional layers, locally computed error kernels transmit signals by applying the convolution of the error kernel with the pre-activation of that layer, aiming to align the forward activations accordingly. In the fully connected case, an error matrix is multiplied by the pre-activation layer to generate targets allowing for the computation of pseudo-gradients that can be used to train the forward activations. While EKDAA is similar to TP in that it creates targets to optimize towards in an encoding/decoding fashion, EK-DAA introduces the novel idea of encoding an error signal with a learned kernel. Similar to FA, EKDAA finds success with using random weights for projecting error signal changing layer-wise dimensionality as weight matrices in the forward and backward pass are asymmetric. In contrast to TP and LRA approaches, with our proposed algorithm, EK-DAA, the forward and backward activities share as minimal information as possible; this helps the model to better overcome poor initialization and stabilization issues. In this section, we describe the forward pass in our notation and then present the details of the EKDAA learning approach.

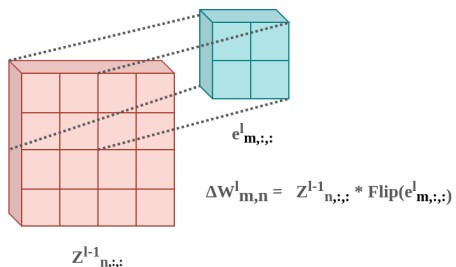

Figure 2: Kernel update to learn the filters $\mathbf{W}^{\ell}_{m,n,:,:}$ with EKDAA. $\mathbf{z}^{\ell-1}_{n,:,:}$, the $n$th post-activation of layer $\ell - 1$, is deconvolved on the $m$th error kernel $e^{\ell}_{m,:,:}$ of layer $\ell$, propagating the error signal to update $\Delta \mathbf{W}^{\ell}_{m,n,:,:}$.

**Notation.** We denote standard convolution with the symbol $*$ and deconvolution with symbol $\circlearrowleft$. Hadamard product is denoted by $\odot$ while $\cdot$ represents a matrix/vector multiplication. $()^T$ denotes the transpose operation. Flip($\mathbf{X}$) is a function for flipping a tensor and is defined as taking the transpose of $\mathbf{X}$ over both the x-axis and y-axis such that the value of an element $\mathbf{X}_{i,j}$ after flipping results in the location $\mathbf{X}_{n-i,n-j}$. Flatten($\mathbf{z}$) means that the input tensor $\mathbf{z}$ is converted to a column vector with a number of rows equal to the number of elements that it originally contained while UnFlatten($\mathbf{z}$) is its inverse (i.e., it converts the vector back to its original tensor shape). We use the notation : to indicate extracting a slice of a certain dimension in a tensor object, i.e., $\mathbf{V}_{j,:,:}$ means that we extract all scalar elements in the $j$th slice of the three dimensional tensor $\mathbf{V}$. Finally, $=$ denotes equality while $\leftarrow$ denotes variable assignment.

**Inference Dynamics.** Given an input (color) image $\mathbf{x}$, inference in a CNN consists of running a feedforward pass through the underlying model, computing the activities (or nonlinear feature

maps) for each layer $\ell$, where the model contains $L_C$ convolutional layers in total. The CNN is parameterized by a set of synaptic tensors $\Theta = \{\mathbf{W}^1, \mathbf{W}^2, ..., \mathbf{W}^{L_C}, \mathbf{W}_y\}$ where the last parameter $\mathbf{W}_y$ is a two-dimensional tensor (or matrix) meant to be used in a softmax/maximum entropy classifier. All other tensors $\mathbf{W}^\ell$, $\ell = 1, 2, ..., L$ are four-dimensional and of shape $\mathbf{W}^\ell \in \mathcal{R}^{N_\ell \times N_{\ell-1} \times h_\ell \times w_\ell}$. This means that any tensor $\mathbf{W}^\ell$ houses $N_\ell$ sets of $N_{\ell-1}$ filters/kernels of shape $h_\ell \times w_\ell$. The bottom tensor $\mathbf{W}^0$, which takes in as input an image, would be of shape $\mathbf{W}^1 \in \mathcal{R}^{N_1 \times N_0 \times h_1 \times w_1}$ where $N_0$ is the number of input color channels, e.g., three, for images of size $h_0 \times w_0$ pixels.

The $m$th feature map of any convolutional layer $\ell$ is calculated as a function of the $N_{\ell-1}$ feature maps of the layer below ($n \in N_{\ell-1}$ – there are $N_{\ell-1}$ input channels to the $m$th channel of layer $\ell$). This is done, with $\mathbf{h}^\ell_{:,:,m}$ initialized as $\mathbf{h}^\ell_{:,:,m} = \mathbf{0}$, in the following manner (bias omitted for clarity):

$$\mathbf{h}^\ell_{m,:,:} \leftarrow \mathbf{h}^\ell_{m,:,:} + \mathbf{W}^\ell_{m,n,:,:} * \mathbf{z}^{\ell-1}_{n,:,:}, \ \forall n \tag{3}$$

$$\mathbf{z}^\ell_{m,:,:} = \phi^\ell(\mathbf{h}^\ell_{m,:,:}) \tag{4}$$

where $\mathbf{W}^\ell_{m,n,:,:}$ denotes the specific filter/kernel that is applied to input channel $n$ when computing values for the $m$th output channel/map. Note that $\phi^\ell$ is the activation function applied to any output channel in layer $\ell$, e.g., $\phi^\ell(v) = \max(0, v)$. Max (or average) pooling is typically applied directly after the nonlinear feature map/channel has been computed, i.e., $\mathbf{z}^\ell_{m,:,:} \leftarrow \Phi_{mp}(\mathbf{z}^\ell_{m,:,:})$.

**Learning Dynamics.** Once inference has been conducted, we may then compute the values needed to adjust the filters themselves. To calculate the updates for each filter in the CNN, EKDAA proceeds in two steps: 1) calculate target activity values for each feature map in each layer (shown in Figure 2) which this is then used to compute the error neuron maps, a type of neuron specialized for computing mismatch signals inspired by predictive processing brain theory (Clark, 2015; Salvatori et al., 2023), and, 2) calculate the adjustments to each filter given the error neuron values. To do so, we introduce a specific set of filter parameters that we call the *error kernels*, each denoted as $\mathbf{E}^\ell_{m,n,:,:}$, for every map and layer in the CNN. This means that including the error kernels as part of the EKDAA-learned CNN parameters yields $\Theta = \{\mathbf{W}^1, \mathbf{E}^1, \mathbf{W}^2, \mathbf{E}^2, ..., \mathbf{W}^{L_C}, \mathbf{E}^{L_C}, \mathbf{W}_y, \mathbf{E}_y\}$. Each error kernel is the same shape as its corresponding convolutional filter, i.e., $\mathbf{E}^\ell \in \mathcal{R}^{N_\ell \times N_{\ell-1} \times h_\ell \times w_\ell}$ (except $\mathbf{E}_y$, which has the same shape as the transpose of $\mathbf{W}_y$).

Assuming that the tensor target activity $\mathbf{y}^\ell$ is available to layer $\ell$, we compute each channel's error neuron map as $\mathbf{e}^\ell_{m,:,:} = -(\mathbf{y}^\ell_{m,:,:} - \mathbf{z}^\ell_{m,:,:})$. Using this mismatch signal, we then work our way down to layer $\ell - 1$ by first convolving this error neuron map to project it downwards, using the appropriate error kernel. Once the projection is complete, if pooling has been applied to the output of each convolutional layer, we then up-sample the projection before computing the final target. This process proceeds formally as follows:

$$\mathbf{e}^\ell_{m,:,:} = -(\mathbf{y}^\ell_{m,:,:} - \mathbf{z}^\ell_{m,:,:}) \tag{5}$$

$$\mathbf{d}^{\ell-1}_{n,:,:} \leftarrow \mathbf{d}^{\ell-1}_{n,:,:} + \mathbf{E}^\ell_{m,n,:,:} \circlearrowleft \mathbf{e}^\ell_{m,:,:}, \ \forall m \in N_\ell \tag{6}$$

$$\mathbf{d}^{\ell-1}_{n,:,:} \leftarrow \Phi_{up}(\mathbf{d}^{\ell-1}_{n,:,:}) \quad // \ (\text{If max-pooling used}) \tag{7}$$

$$\mathbf{y}^{\ell-1}_{n,:,:} = \phi^{\ell-1}(\mathbf{h}^{\ell-1}_{n,:,:} - \beta \mathbf{d}^{\ell-1}_{n,:,:}) \tag{8}$$

where we see that $\Phi_{up}()$ denotes the up-sampling operation (to recover the dimensionality of the map before max-pooling was applied). If pooling was not used in layer $\ell - 1$, then Equation 7 is omitted in the calculation of layer $\ell - 1$'s target activity. Notice that the update rule has a recursive nature; it requires the existence of $\mathbf{y}^\ell$ which in turn would have been created by applying Equations 5-8 to the layer above, $\ell + 1$. Thus, the base case target activity $\mathbf{y}^L$, which would exist at the very

---

**Algorithm 1** EKDAA for a CNN with max-pooling and fully-connected maximum entropy output.

// Feedforward inference
**Input:** sample $(\mathbf{y}, \mathbf{x})$ and $\Theta$
**function** INFER$(\mathbf{x}, \Theta)$
   // Pass data thru convolution stack
   // Get image input channels
   $\mathbf{z}^0_{n,:,:} = \mathbf{x}_{n,:,:}, \ \forall n \in N_0$
   **for** $\ell = 1$ to $L_C$ **do**
      // Calculate feature maps for layer $\ell$
      $\mathbf{h}^\ell_{m,:,:} = \mathbf{0}, \forall m \in N_\ell$
      **for** $m = 1$ to $N_\ell$ **do**
         $\mathbf{h}^\ell_{m,:,:} \leftarrow \mathbf{h}^\ell_{m,:,:} + \mathbf{W}^\ell_{m,n,:,:} * \mathbf{z}^{\ell-1}_{n,:,:}, \ \forall n$
         $\mathbf{z}^\ell_{m,:,:} = \phi^\ell(\mathbf{h}^\ell_{m,:,:}),$
         $\mathbf{z}^\ell_{m,:,:} \leftarrow \Phi_{mp}(\mathbf{z}^\ell_{m,:,:})$
   $\mathbf{h}_y = \mathbf{W}_y \cdot \text{Flatten}(\mathbf{z}^{L_C}), \ \mathbf{z}_y = \sigma(\mathbf{h}_y)$
   $\Lambda = \{(\mathbf{h}^1, ..., \mathbf{h}^{L_C}, \mathbf{h}_y), (\mathbf{z}^0, ..., \mathbf{z}^{L_C}, \mathbf{z}_y)\}$
   **Return** $\Lambda$

// Calculate weight updates via EKDAA
**Input:** Statistics $\Lambda$, target $\mathbf{y}$, $\beta$, and $\Theta$
**function** CALCUPDATES$(\Lambda, \mathbf{y}, \Theta)$
   $\mathbf{h}^1, ..., \mathbf{h}^{L_C}, \mathbf{h}_y, \mathbf{z}^0, ..., \mathbf{z}^{L_C}, \mathbf{z}_y \leftarrow \Lambda, \ \mathbf{y}^L = \mathbf{y}$
   // Compute softmax weight updates
   $\mathbf{e}_y = -(\mathbf{y} - \mathbf{z}_y)$
   $\Delta \mathbf{W}_y = \mathbf{e}_y \cdot \left(\text{Flatten}(\mathbf{z}^{L_C})\right)^T,$
   $\Delta \mathbf{E}_y = -\gamma(\Delta \mathbf{W}_y)^T$
   $\mathbf{y}^{L_C} = \phi^{L_C}\left(\text{Flatten}(\mathbf{h}^{L_C}) - \beta(\mathbf{E} \cdot \mathbf{e}_y)\right)$
   // Compute convolutional kernel updates
   $\mathbf{y}^{L_C} \leftarrow \text{UnFlatten}(\mathbf{y}^{L_C})$
   **for** $\ell = L_C$ to $1$ **do**
      **for** $m = 1$ to $N_\ell$ **do**
         $\mathbf{e}^\ell_{m,:,:} = -(\mathbf{y}^\ell_{m,:,:} - \mathbf{z}^\ell_{m,:,:})$
      $\mathbf{d}^\ell = \mathbf{0}, \forall n \in N_{\ell-1}$
      **for** $n = 1$ to $N_{\ell-1}$ **do**
         **for** $m = 1$ to $N_\ell$ **do**
            $\mathbf{d}^{\ell-1}_{n,:,:} \leftarrow \mathbf{d}^{\ell-1}_{n,:,:} +$
               $(\mathbf{E}^\ell_{m,n,:,:} \circlearrowleft \mathbf{e}^\ell_{m,:,:})$
         $\mathbf{y}^{\ell-1}_{n,:,:} = \phi^{\ell-1}(\mathbf{h}^{\ell-1}_{n,:,:} - \beta \Phi_{up}(\mathbf{d}^{\ell-1}_{n,:,:}))$
      **for** $m = 1$ to $N_\ell$ **do**
         **for** $n = 1$ to $N_{\ell-1}$ **do**
            $\Delta \mathbf{W}^\ell_{m,n,:,:} = \mathbf{z}^{\ell-1}_{n,:,:} * \text{Flip}(\mathbf{e}^\ell_{m,:,:})$
            $\Delta \mathbf{E}^\ell_{m,n,:,:} = -\gamma(\Delta \mathbf{W}^\ell_{m,n,:,:})^T$
   $\Delta = \{\Delta \mathbf{W}^0, \Delta \mathbf{E}^0, ..., \mathbf{W}^{L_C}, \Delta \mathbf{E}^{L_C}, \mathbf{W}_y, \mathbf{E}_y\}$
   **Return** $\Delta$

---

top (or highest level) of the CNN, and, in the case of supervised classification, which is the focus of this paper, this would be the target label vector $\mathbf{y}$ associated with input image $\mathbf{x}$.

Once targets have been computed for each convolutional layer, the adjustment for each kernel in each layer requires a specialized local rule that entails convolving the post-activation maps of the level below with the error neuron map at $\ell$. Formally, this means:

$$\Delta \mathbf{W}^\ell_{m,n,:,:} = \mathbf{z}^{\ell-1}_{n,:,:} * \text{Flip}(\mathbf{e}^\ell_{m,:,:}) \tag{9}$$

$$\Delta \mathbf{E}^\ell_{m,n,:,:} = -\gamma(\Delta \mathbf{W}^\ell_{m,n,:,:})^T \tag{10}$$

which can then subsequently be treated as the gradient to be used in either a stochastic gradient descent update, i.e., $\mathbf{W}^\ell_{m,n,:,:} \leftarrow \mathbf{W}^\ell_{m,n,:,:} - \lambda \Delta \mathbf{W}^\ell_{m,n,:,:}$, or a more advanced rule such as Adam (Kingma & Ba, 2017) or RMSprop (Tieleman & Hinton, 2012).

In Algorithm 1, we provide a mathematical description of how EKDAA would be applied to a deep CNN specialized for classification. Note that, while this paper focuses on feedforward classification, our approach is not dependent on the type of task that the CNN is required to solve. For example, one could readily employ our approach to construct unsupervised convolutional autoencoders, to craft alternative convolutional architectures that solve other types of computer vision problems, e.g., image segmentation, or to build complex models that process time series information, i.e., temporal/recurrent convolutional networks. One key advantage of the above approach is that the

---

**Algorithm 2** EKDAA for fully-connected and maximum entropy output layers.

// Feedforward inference
**Input:** sample $(\mathbf{y}, \mathbf{x})$ and $\Theta$
**function** INFER$(\mathbf{x}, \Theta)$
    $\mathbf{z}^0 = \mathbf{x}$         ▷ $\mathbf{x}$ could be Flatten$(\mathbf{z}^{L_C})$
    // Calculate fully-connected layers
    **for** $\ell = 1$ to $L_{FC}$ **do**
        $\mathbf{h}^\ell = \mathbf{W}^\ell \cdot \mathbf{z}^{\ell-1}$
        $\mathbf{z}^\ell = \phi^\ell(\mathbf{h}^\ell)$,
    // Calculate softmax outputs
    $\mathbf{h}_y = \mathbf{W}_y \cdot (\mathbf{z}^{L_{FC}})$, $\mathbf{z}_y = \sigma(\mathbf{h}_y)$
    $\Lambda = \{(\mathbf{h}^1, ..., \mathbf{h}^{L_{FC}}, \mathbf{h}_y), (\mathbf{z}^0, ..., \mathbf{z}^{L_{FC}}, \mathbf{z}_y)\}$
    **Return** $\Lambda$

// Calculate weight updates via EKDAA
**Input:** Statistics $\Lambda$, target $\mathbf{y}$, $\beta$, and $\Theta$
**function** CALCUPDATES$(\Lambda, \mathbf{y}, \Theta)$
    $\mathbf{h}^1, ..., \mathbf{h}^{L_{FC}}, \mathbf{h}_y, \mathbf{z}^0, ..., \mathbf{z}^{L_{FC}}, \mathbf{z}_y \leftarrow \Lambda$, $\mathbf{y}^L = \mathbf{y}$
    // Compute softmax weight updates
    $\mathbf{e}_y = -(\mathbf{y} - \mathbf{z}_y)$
    $\Delta\mathbf{W}_y = \mathbf{e}_y \cdot (\mathbf{z}^{L_{FC}})^T$, $\Delta\mathbf{E}_y = -\gamma(\Delta\mathbf{W}_y)^T$
    $\mathbf{y}^{L_{FC}} = \phi^{L_{FC}}(\mathbf{h}^{L_{FC}} - \beta(\mathbf{E} \cdot \mathbf{e}_y))$
    // Compute fully-connected weight updates
    **for** $\ell = L_{FC}$ to $1$ **do**
        $\mathbf{e}^\ell = -(\mathbf{y}^\ell - \mathbf{z}^\ell)$
        $\mathbf{y}^{\ell-1} = \phi^{\ell-1}\Big(\mathbf{h}^{\ell-1} - \beta\big((\mathbf{E}^\ell \cdot \mathbf{e}^\ell)\big)\Big)$
        $\Delta\mathbf{W}^\ell = \mathbf{e}^\ell \cdot (\mathbf{z}^{\ell-1})^T$, $\Delta\mathbf{E}^\ell = -\gamma(\Delta\mathbf{W}^\ell)^T$
    $\Delta = \{\Delta\mathbf{W}^0, \Delta\mathbf{E}^0, ..., \mathbf{W}^{L_C}, \Delta\mathbf{E}^{L_C}, \mathbf{W}_y, \mathbf{E}_y\}$
    **Return** $\Delta$

---

test-time inference of an EKDAA-trained CNN is no slower than a standard backprop-trained CNN, given that the forward pass computations remain the same. EKDAA also benefits from model stabilization in comparison to BP. Even if the incoming pre-activations contain extreme values due to poor initialization, EKDAA credit assignment will still give usable error signals to learn from; this is due to the fact that such signals merely represent difference between target and actual activities (subtractive Hebbian learning) rather than differentiable values as in backprop.

**Fully-Connected Synaptic Updates.** As made clear before, EKDAA trains systems composed of convolution/pooling operators (Algorithm 1); however, it can also be used to train fully/densely-connected transformations. Specifically, for fully-connected layers, EKDAA recovers an LRA-based scheme Ororbia & Mali (2019) as shown in Algorithm 2, i.e., EKDAA is a generalization of LRA (LRA was developed for fully-connected layers). In general, like backprop, EKDAA converges to loss function minima because its pseudo-gradients, which are a function of the error signals, satisfy the condition that for a given weight matrix, $W_n$: $\Delta W_n \leq |\Delta\tilde{W}_n - 90°|$, where $\Delta\tilde{W}_n$ is the exact calculated derivative for matrix $W_n$. Although the approximate gradients are almost never equal to backprop's exact gradients, they are always within 90 degrees and, thus, they tend towards the direction of backprop's (steepest) gradient descent. Note that EKDAA's pseudo-gradients do not take steps towards the loss surface minima as greedily as backprop's gradients do; however, over many training iterations, they converge to loss surface points similar to those found by backprop.

## 4 Experimental Setup and Results

### 4.1 Datasets and Experimental Tasks

To understand learning capacity for model fitting and generalization under EKDAA, we design and train several models and test them against three standard datasets, Fashion MNIST (Xiao et al., 2017) (FMNIST), CIFAR-10 (Krizhevsky et al., 2014), and SVHN (Netzer et al., 2011).

Fashion MNIST, while only being a single channel (gray-scale) image dataset at a $[28 \times 28]$ resolution, has a more complicated pixel input space than MNIST, facilitating a better analysis of the convolutional contribution to overall network performance. SVHN and CIFAR-10 images are

| Dataset/Algorithm | Memory Required (MB) | Computation Time (Sec) |
|---|---|---|
| FMNIST/BP | 1517.29 | 96.36 +/- 1.79 |
| FMNIST/EKDAA | 2574.25 | 96.39 +/- 9.16 |
| SVHN/BP | 4723.84 | 316.84 +/- 19.91 |
| SVHN/EKDAA | 4728.03 | 268.95 +/- 54.56 |
| CIFAR-10/BP | 4723.84 | 319.72 +/- 13.59 |
| CIFAR-10/EKDAA | 4728.03 | 271.99 +/- 11.58 |

Table 1: Memory and computation time required for backprop and EKDAA models. GPU memory and average computation time with variance is recorded per 1000 updates with 10 trial runs.

of shape $[32 \times 32]$ pixels and are represented in three color channels, containing more complicated patterns and motifs compared to the gray scale Fashion MNIST (SVHN even contains many images with distractors at the sides of the character that the image is centered on). Fully-connected layers are not strong enough to learn the spatial relationships between pixels, and, as a result, convolutional filters will be required to learn the key patterns within the data that would help in distinguishing between the various classes.

Taken together, the FMNIST, CIFAR-10, and SVHN datasets allow us to study how well EKDAA learns filters (when engaged in the process of data fitting) and also how effective it is in creating models that generalize on visual datasets. Additionally, we show that our networks can be trained using non-differentiable activations, such as the signum function, or, more formally: $\text{signum}(x) = 1$ if $x > 0, 0$ if $x = 0, -1$ if $x < 0$. In this case, we train all convolutional and fully-connected layers using signum as the activation function, except for the softmax output layer, in order to investigate how well an EKDAA-driven network handles non-differentiable activity without specific tuning.

**Technical Implementation.** We design CNNs for the Fashion MNIST, CIFAR-10, and SVHN datasets. The FMNIST CNN consists of three convolutional layers before flattening and propagating through one fully-connected softmax layer. The filter size is $[3 \times 3]$ for all convolutional layers with the first layer starting with one channel, expanding to 32, then to 64, and ending at 128 filters. The fully-connected layers start after flattening the filters, which are then propagated through 128 fully-connected nodes before ending at 10 outputs (one per image class). Max-pooling with a kernel of $[2 \times 2]$ and a stride of 2 was used at the end of the first and second layers of convolution.

The CIFAR-10 and SVHN models use six layers of convolution and are inspired by the blocks of convolution and pooling layers used in the VGG family of networks (Simonyan & Zisserman, 2014). First, two convolutional layers are used before finally passing through a max-pooling layer with a kernel of $[2 \times 2]$ and a stride of two. Three of these mini-blocks of two convolution layers, followed by a max-pooling layer, are used to build the final network. The first three layers of convolution use 64 filters while the last three layers use 128 filters. All layers use a filter size of $[3 \times 3]$. After traversing through the last convolutional layer, the final neural activities are flattened and propagated through a single 128-node, fully-connected layer before shrinking down to 10 output nodes (which are subsequently run through a softmax nonlinearity). Both Fashion MNIST, SVHN, and CIFAR-10 models use a very small amount of fully-connected nodes and instead use multiple large filter layers to learn and extract distributed representations (see Appendix for details).

Each model was tuned for optimal performance and several hyper-parameters were adjusted, i.e., batch size, learning rate, filter size, number of filters per layer, number of fully-connected nodes per layer, weight initialization, optimizer choice, and dropout rate (details can be found in Appendix). The same architecture was used for both the EKDAA model and the other learning mechanisms that it was compared against. Models were optimized using stochastic gradient descent with momentum and Pascanu re-scaling (Pascanu et al., 2013) was applied to all layer-wise weight updates.

| | FMNIST | | SVHN | | CIFAR-10 | |
|---|---|---|---|---|---|---|
| | **Train Acc** | **Test Acc** | **Train Acc** | **Test Acc** | **Train Acc** | **Test Acc** |
| BP | $95.31 \pm 0.18$ | $89.97 \pm 0.14$ | $90.98 \pm 0.23$ | $88.52 \pm 0.10$ | $83.33 \pm 0.22$ | $71.08 \pm 0.08$ |
| BP (FC) | $92.91 \pm 0.39$ | $87.02 \pm 0.41$ | $84.36 \pm 0.12$ | $79.81 \pm 0.22$ | $57.05 \pm 0.34$ | $55.03 \pm 0.29$ |
| LRA-E (FC) | $93.59 \pm 0.26$ | $87.58 \pm 0.33$ | $80.17 \pm 0.08$ | $73.24 \pm 0.19$ | $58.10 \pm 0.28$ | $55.51 \pm 0.42$ |
| EKDAA | $95.83 \pm 0.33$ | $90.01 \pm 0.11$ | $84.31 \pm 0.21$ | $82.27 \pm 0.19$ | $75.05 \pm 0.27$ | $63.38 \pm 0.12$ |
| EKDAA, Sig. | $94.00 \pm 0.14$ | $88.69 \pm 0.06$ | $79.43 \pm 0.09$ | $76.87 \pm 0.13$ | $64.22 \pm 0.12$ | $59.71 \pm 0.08$ |
| HSIC (Ma et al., 2019) | | $88.30 \pm --$ | | | | $59.50 \pm --$ |
| FA | $95.30 \pm 0.51$ | $89.10 \pm 0.18$ | $79.18 \pm 00.22$ | $76.50 \pm 00.18$ | $77.50 \pm 0.25$ | $58.80 \pm 0.11$ |
| DFA | $93.99 \pm 0.32$ | $88.90 \pm 0.10$ | $82.50 \pm 00.24$ | $80.30 \pm 00.21$ | $79.50 \pm 0.20$ | $60.50 \pm 0.08$ |
| SDFA | $94.10 \pm 0.28$ | $89.00 \pm 0.10$ | $84.50 \pm 00.23$ | $81.40 \pm 00.19$ | $80.00 \pm 0.19$ | $59.60 \pm 0.06$ |
| DRTP | $93.50 \pm 0.40$ | $87.99 \pm 0.15$ | $85.21 \pm 00.21$ | $81.90 \pm 00.20$ | $79.50 \pm 0.22$ | $58.20 \pm 0.14$ |

Table 2: Train and test accuracy on the selected datasets. Mean and standard deviation over 10 trials reported. **Note**: The signum (sig.) function is included to demonstrate that we are able to successfully train a non-differentiable activation with EKDAA and obtain reasonable performance. BP results are shown to serve as a best case scenario when training with optimal gradients.

All models were trained on the original datasets, at the original resolutions, without any augmentation or pre-training. Unlike backprop, EKDAA did not benefit from extensive heuristic knowledge on optimal parameterization, making a grid search for parameters ineffective as the hyperparameter tuning limits would be significantly wider than with backprop. As a result, for tuning, the learning rate was tuned from the range $1e-1$ to $1e-4$, number of filters were tuned from the range 32 to 128, dropout was tuned from 0 to 0.5, and the activation function was evaluated to be either the hyperbolic tangent (tanh) or the linear rectifier (relu). The final meta-parameter setting for EKDAA was a learning rate of $0.5e-3$, 0.9 momentum rate, tanh for activation, and a dropout rate of 0.1 for filters and 0.3 for fully-connected layers (complete model specifications are in the Appendix). All model weights were randomly initialized with system time as a seed. For EKDAA, the error kernels were randomly initialized with the Glorot uniform weight initialization scheme (Glorot & Bengio, 2010). Furthermore, all models were trained on a single Tesla P4 GPU with 8GB of GPU RAM and ran on Linux Ubuntu 18.04.5 LTS, using Tensorflow 2.1.0. The code for this work has been designed in a novel library that allows for defining convolutional and fully-connected models with the ability to quickly change the learning mechanism between BP and EKDAA. The library also allows for defining new custom learning rules for analysis. While this codebase offers an advantage for analyzing learning mechanisms, it has been custom written without the optimization techniques that common libraries have implemented. This codebase takes advantage of Tensorflow tensors when possible but has a custom defined forward and backward pass that is not nearly as memory or computationally efficient as it could be. Our library implementation can be found at: *https://github.com/tzee/EKDAA-Release*.

In addition to train and test accuracy, we compare the GPU memory usage and computation time required to train EKDAA and backprop. Results for this are shown in Table 1. Overall, EKDAA shows a slight improvement in computation time as models become larger and, overall, only results in a minor increase in required memory to run the same model than backprop. Overall, EKDAA's architecture does not exhibit worse computational requirements than those of backprop.

## 4.2 Results and Discussion

We analyzed EKDAA by comparing it to several bio-inspired learning schemes such as HSIC (Ma et al., 2019), RFA (Lillicrap et al., 2016), DFA (Nøkland, 2016), sparse direct feedback alignment (SDFA) (Crafton et al., 2019), and direct random target projection (DRTP) (Frenkel et al., 2021) (see Appendix for baseline details). The results are presented in Table 2 (we also added two fully-

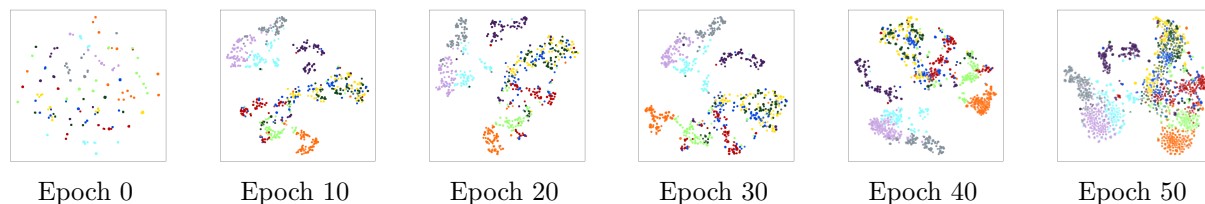

| Epoch 0 | Epoch 10 | Epoch 20 | Epoch 30 | Epoch 40 | Epoch 50 |

Figure 3: t-SNE visualization depicting the learned representations of EKDAA, shown for Epoch 0 (initial weights) through and up to the final training epoch for FMNIST.

connected baselines – an MLP trained by backprop and one trained by LRA-E (Ororbia & Mali, 2019)). Comparable BP results are shown only to provide intuition on how well the constructed models could perform if trained with precise gradients. We find EKDAA performs competitively with the other algorithms and exhibits both increased train and test accuracy on the natural color images, i.e., SVHN and CIFAR-10. Additionally, when testing EKDAA with the signum activation, we find the resulting CNN operates with the non-differentiable function successfully on FMNIST.

Figure 3 shows the t-SNE plots of the outputs of the last layer of a CNN across epochs, to demonstrate how EKDAA disentangles the feature space of Fashion MNIST. The t-SNE plots were generated every 10th epoch and visualized using default t-SNE parameters (i.e. no tuning) with a perplexity value of 30 and the maximum number of iterations set to 1000. Qualitatively, we find EKDAA successfully learns to group features together with primarily convolutional layers, indicating the error kernels learned are indeed benefiting the CNN. Figure 4 shows the train and test learning curves of BP and EKDAA (plotting accuracy as a function of epoch for the best configuration of the model using each algorithm).

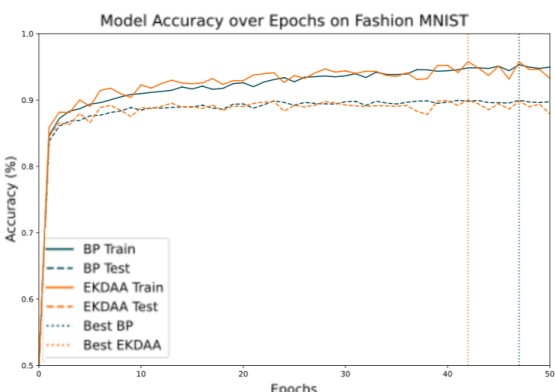

Figure 4: FMNIST train and test accuracy curves for BP and EKDAA.

While many biologically-plausible alternatives have also been developed to learn models of natural images, many of them incorporate error derivatives as part of the process and the architectures they are generally applied to have been designed with multiple, large fully-connected layers with only a few convolutional layers. We argue adding many fully-connected layers corrupts the input signal such that the model is engaged in a greedy optimization resulting in fitting to noise rather than extracting useful features. The role of convolution in such models is still debatable and, as a result, it is difficult to determine if model performance results from the bio-plausible learning mechanism or from the fully-connected layers. In contrast, EKDAA emphasizes the role of convolutional filters in extracting useful features while reducing fully-connected elements. Our results on the three datasets examined above validate that this approach still yields models that generalize well.

**Limitations.** The main limitation of the proposed EKDAA algorithm is currently its scalability to massive datasets, especially when compared with highly optimized tensor computations implemented in standard deep learning libraries that support backprop-based convolution/deconvolution operations. Due to the current lack of advanced optimizations compared to frameworks such as TensorFlow and PyTorch, supported by large tech companies, EKDAA is not as efficient, thus requiring more computational resources than the established frameworks. Based on our practical experience with our custom library that implements EKDAA, it does not scale easily to very large

networks with many filters given our constrained computational budget and hardware. Specifically, we used one Ubuntu 18.04 server with 8GB Tesla P4, an Intel Xeon CPU E5-2650, and 256GB of RAM. Future work includes further modularizing our EKDAA library, focusing to improve the algorithm's ability to scale to datasets like ImageNet (Deng et al., 2009), despite limited resources.

**Broader Impacts.** Backprop generally works well on a carefully parameterized network, yet it has a few drawbacks. For example, it requires computing gradients layer-by-layer, enforcing strict requirements on information flow, i.e. needing to propagate in only a feed-forward fashion (lateral connections make no sense for backprop). Local learning mechanisms do not have these restrictions, facilitating the exploration of novel network structures and exotic forms of propagation flow.

We introduce a novel framework for training images without the need for backprop. While our proposed scheme is limited with scale and speed compared to the highly optimized tensor computations implemented in modern deep learning libraries, this work serves as a foundation for exploring local learning for images. We introduce EKDAA, which learns error kernels from local layers to better represent image data in a backprop-free manner. While various bio-plausible methodologies have been developed in recent years, their learning rules tend to focus on fully-connected layers. Some methods use convolution, but often fix randomly initialized filters or use backprop to train those layers. With learnable error kernels, we introduce a way to transfer error signals through a model's convolutional layers while not requiring gradient information like backprop. EKDAA updates are local and circumvent the weight transport problem. Continued work in this area may have profound impacts on future model development by alleviating backprop's severe restrictions while still providing the ability to model highly complex, high-dimensional data such as natural color images.

## 5 Conclusion

We have presented an initial exploration of a back-propagation-free convolutional neural network learning algorithm. We implemented a local feedback mechanism that transmits information across layers to compute target activity values and relevant error neuron maps (independent of activation function type), resulting in Hebbian-like update rules for the convolutional filters of a CNN. Specifically, this credit assignment process was made possible by introducing a mechanism that we call the error kernel, which provides a means to reverse filter error neuron activity signals and complements the normal filters used to extract features in convolutional models. We refer to our proposed process as the *error-kernel driven activation alignment* (EKDAA) method.

We compared various algorithms in training a small CNN and found that EKDAA outperforms other bio-inspired alternatives on natural color images for classification on the datasets tested. Notably, our method offers several benefits: 1) it resolves the major bio-implausibility of the weight transport problem, 2) works with non-differentiable activities, and 3) is relatively computationally efficient since it can operate in a layer-wise parallel/asynchronous fashion. Our experiments demonstrate that EKDAA learns "good" representations during training and, furthermore, we found that an EKDAA-trained CNN acquires latent representations that improve over time (epochs) as training evolves. In addition, we find that EKDAA has similar computational and memory requirements as backprop, while exhibiting similar convergence behavior. While there is still much to explore in future work, we have successfully presented an analysis of the novel EKDAA algorithm, yielding promising evidence that it can train convolutional networks without backprop. Future directions to expand this work involve expanding to larger, more complex architectures and imaging data, as well as a study of the properties a Hebbian-based algorithm like EKDAA offer over backprop. The implication of this could have far-reaching effects in the future designs of CNN architectures.

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

## Appendix

## A  Experimental Setup Details

We performed a grid search for all of the models investigated in this work in order to find optimal meta-parameters and extract optimal behavior for each. Primarily, tuned hyper-parameters included: batch size, learning rate, filter size, number of filters per layer, number of fully-connected

nodes/units per layer, weight initialization, choice of optimizer, and the dropout rate. Note that this work does not aim to obtain state-of-the-art image classification results. Rather, its intent is to present a method that efficiently tackles the credit assignment issues in a convolution neural network (CNN) by effectively operating with our proposed error kernel mechanism. Furthermore, our method offers additional flexibility in design choices (such as permitting the use of non-differentiable activation functions).

**Meta-parameter Tuning:** We report our grid search ranges for each model's meta-parameters in Tables 2 (EKDAA), 7 (DFA), 6 (FA), 9 (RDFA), and 8 (SDFA), respectively. Furthermore, in the "Best" column, we report the final values selected/used for the models reported in the main paper.

**Architecture Design:** In Table 4, we present the architectures used across the learning algorithms investigated in this paper, i.e., the proposed EKDAA, feedback alignment (FA, also referred to as RFA in the main paper), direct feedback alignment (DFA), sparse direct feedback alignment (SDFA), and random direct feedback alignment (RDFA). We built the models for Fashion MNIST, SVHN, and CIFAR-10 to include several layers of convolution (conv), with a sizeable amount of filters, and only small (in terms of dimensionality) fully-connected (fc) layers to focus the learning process on adapting/using the model kernels/filters to extract useful features from the input image signals. In particular, the model for SVHN and CIFAR-10 had multiple layers with 128 filters per layer and, before flattening the activities for the fully-connected layers, the image size was reduced using three max pooling layers in order to propagate forward the image to obtain a $[4 \times 4]$ resolution.

**General Comments/Discussion:** With respect to the main paper's results, what is significant about our findings is that EKDAA demonstrates that adjusting the synaptic weight parameters of a CNN is possible using recurrent error synapses formulated as error kernels themselves. This means that the target feature map values (and the error neuron maps that calculate the distance between the original feature maps and these targets) inherent to our backprop-free computational process provide useful teaching signals that facilitate the learning of useful neural vision architectures. The main results of our paper provide promising initial evidence that EKDAA can serve as a potentially useful bio-inspired alternative to backprop for training CNNs on natural images.

## B  Asset Usage

We build our codebase on top of TensorFlow 2.0 for fundamental functionality. TensorFlow is open-source with an Apache license. In addition, for analysis we use the publicly available Fashion-MNIST, SVHN, and CIFAR-10 datasets, all of which have licenses permitting unlimited use and modification. In addition, none of the datasets used in this study entail any data that could be considered sensitive (thus not requiring data consent) or offensive.

## C    Notation

| Variable | Definition |
|---|---|
| $X_l$ | input to layer $l$ |
| $\Delta X_l$ | error signal from the output of layer $l$ |
| $Y$ | ground truth corresponding to $X$ |
| $H_l$ | pre-activation of layer $l$ |
| $Z_l$ | post-activation for layer $l$ |
| $W_l$ | weights for layer $l$ |
| $W_{m,n,:,:}^l$ | weights for a layer $l$ with feature map $[m,n]$ |
| $\Delta W_l$ | weight updates for layer $l$ |
| $\mathbf{V_{j,:,:}}$ | extract all scalar elements in the $j^{th}$ slice of the $2^{nd}$ order Tensor $\mathbf{V}$ |
| $e_l$ | convolutional error kernels for layer $l$ |
| $\Delta e_l$ | error signal update for error kernels for layer $l$ |
| $E_l$ | error weights or matrix for layer $l$ |
| $\Delta E_l$ | error signal update for error weights or matrix for layer $l$ |
| $\lambda$ | learning rate for learnable parameter updates |
| $\Theta$ | complete set of trainable parameters $\{W_0 : W_l, E_0 : E_l, e_0 : e_l\}$ |
| $\Lambda$ | complete set of pre- and post- activations $\{h_0 : h_l, z_0 : z_l\}$ |
| $\beta$ | tuneable parameter to control error signal strength through layers |
| $\gamma$ | tuneable parameter to control error signal strength for $\Delta E$ |
| Operator | Definition |
| $=$ | equality |
| $\leftarrow$ | variable assignment |
| $\lvert X \rvert$ | absolute value of X |
| $:$ | a slice of a tensor object |
| $*$ | convolution operator |
| $\circlearrowleft$ | deconvolution operator |
| $\odot$ | Hadamard product |
| $\cdot$ | matrix/vector multiplication |
| $()^T$ | transpose |
| $\phi$ | activation function |
| $\sigma$ | softmax activator |
| $\Phi_{mp}$ | max pooling operator |
| $\Phi_{up}$ | up-sampling to pre-pooling size |
| $Flip(X)$ | transpose of $X$ across $x$ and $y$ axis |
| $Flatten(X)$ | X is converted to an equivalent column vector |
| $UnFlatten(X)$ | X is reshaped into its pre-flattened shape |

Table 3: Table of all notations with their respective definitions.

## D    Model and Training Specifications

| Layer | Fashion MNIST | Fashion MNIST Output | SVHN/CIFAR-10 | SVHN/CIFAR-10 Output |
|---|---|---|---|---|
| L0 | Input | [:, 28, 28, 1] | Input | [:, 32, 32, 3] |
| L1 | Conv1 (1, 32) [3 x 3] | [:, 28, 28, 32] | Conv1 (3, 64) [3 x 3] | [:, 32, 32, 64] |
| L2 | MaxP1 (2, 2) | [:, 14, 14, 32] | Conv2 (64, 64) [3 x 3] | [:, 32, 32, 64] |
| L3 | Conv2 (32, 64) [3 x 3] | [:, 14, 14, 64] | MaxP1 (2, 2) | [:, 16, 16, 64] |
| L4 | MaxP2 (2, 2) | [:, 7, 7, 64] | Conv3 (64, 64) [3 x 3] | [:, 16, 16, 64] |
| L5 | Conv3 (64, 128) [3 x 3] | [:, 7, 7, 128] | Conv4 (64, 128) [3 x 3] | [:, 16, 16, 128] |
| L6 | Flatten() | [:, 6272] | MaxP2 (2, 2) | [:, 8, 8, 128] |
| L7 | FC1 (6272, 128) | [:, 128] | Conv5 (128, 128) [3 x 3] | [:, 8, 8, 128] |
| L8 | Softmax (128, 10) | [:, 10] | Conv6 (128, 128) [3 x 3] | [:, 8, 8, 128] |
| L9 | - | - | MaxP3 (2, 2) | [:, 4, 4, 128] |
| L10 | - | - | Flatten() | [:, 2048] |
| L11 | - | - | FC1 (2048, 128) | [:, 128] |
| L12 | - | - | Softmax (128, 10) | [:, 10] |

Table 4: Model architectures that were trained on Fashion MNIST, SVHN, and CIFAR-10. The layers of each model are defined as well as the outputs from each layer.

| Parameter | Range Min | Range Max | Interval | Activation Functions | Best |
|---|---|---|---|---|---|
| batch_size | 50 | 250 | 50 | - | 50 |
| learning_rate | 1e-5 | 1e-2 | 0.5 | - | 5e-4 |
| filter_size | 3 | 7 | 2 | - | 3 |
| num_filters | 32 | 256 | 32 | - | - |
| fc_per_layer | 128 | 128 | - | - | 128 |
| weight_init | - | - | - | glorot_uniform, glorot_normal | glorot_uniform |
| optimizer | - | - | - | tanh, relu, signum | tanh |
| dropout | 0.0 | 0.5 | 0.1 | - | 0.1 conv, 0.3 fc |

Table 5: Hyper-parameter tuning ranges and best found parameters for EKDAA.

| Parameter | Range Min | Range Max | Increment | Activation Functions | Best |
|---|---|---|---|---|---|
| batch_size | 32 | 256 | 64 | - | 64 |
| learning_rate | 5e-5 | 3e-2 | 0.5 | - | 5e-4 |
| filter_size | 3 | 7 | 2 | - | 3 |
| num_filters | 32 | 256 | 32 | - | - |
| fc_per_layer | 128 | 128 | - | - | 128 |
| weight_init | - | - | - | glorot_uniform, glorot_normal | glorot_normal |
| optimizer | - | - | - | tanh, relu | relu |
| dropout | 0.0 | 0.5 | 0.1 | - | 0.1 conv, 0.3 fc |

Table 6: Hyper-parameter tuning ranges and best found parameters for FA.

| Parameter | Range Min | Range Max | Increment | Activation Functions | Best |
|---|---|---|---|---|---|
| batch_size | 32 | 256 | 64 | - | 64 |
| learning_rate | 5e-5 | 3e-2 | 0.5 | - | 5e-3 |
| filter_size | 3 | 7 | 2 | - | 3 |
| num_filters | 32 | 256 | 32 | - | - |
| fc_per_layer | 128 | 128 | - | - | 128 |
| weight_init | - | - | - | glorot_uniform, glorot_normal | glorot_uniform |
| optimizer | - | - | - | tanh, relu | relu |
| dropout | 0.0 | 0.5 | 0.1 | - | 0.1 conv, 0.2 fc |

Table 7: Hyper-parameter tuning ranges and best found parameters for DFA.

| Parameter | Range Min | Range Max | Increment | functions | Best |
|---|---|---|---|---|---|
| batch_size | 32 | 256 | 64 | - | 64 |
| learning_rate | 5e-5 | 3e-2 | 0.5 | - | 3e-3 |
| filter_size | 3 | 7 | 2 | - | 3 |
| num_filters | 32 | 256 | 32 | - | - |
| fc_per_layer | 128 | 128 | - | - | 128 |
| weight_init | - | - | - | glorot_uniform, glorot_normal | glorot_uniform |
| optimizer | - | - | - | tanh, relu | tanh |
| dropout | 0.0 | 0.5 | 0.1 | - | 0.1 conv, 0.1 fc |

Table 8: Hyper-parameter tuning ranges and best found parameters for SDFA.

| Parameter | Range Min | Range Max | Increment | functions | Best |
|---|---|---|---|---|---|
| batch_size | 32 | 256 | 32 | - | 32 |
| learning_rate | 5e-5 | 3e-2 | 0.5 | - | 4e-3 |
| filter_size | 3 | 7 | 2 | - | 3 |
| num_filters | 32 | 256 | 32 | - | - |
| fc_per_layer | 128 | 128 | - | - | 128 |
| weight_init | - | - | - | glorot_uniform, glorot_normal | glorot_normal |
| optimizer | - | - | - | tanh, relu | relu |
| dropout | 0.0 | 0.5 | 0.1 | - | 0.2 conv, 0.3 fc |

Table 9: Hyper-parameter tuning ranges and best found parameters for RDFA.

