# OpenReview forum: "A Robust Backpropagation-Free Framework for Images"
_TMLR — Accepted by TMLR_

### Review · Reviewer_B7BR · 2023-07-03

**Summary Of Contributions:**

This paper presents the Error-Kernel Driven Activation Alignment (EKDAA) algorithm, an approach aimed at solving the weight transport problem in deep learning algorithms. The weight transport problem refers to the fact that, in current deep learning algorithms, the computation of gradients via backpropagation requires knowledge of feed-forward activities. The EKDAA overcomes this issue by utilizing locally derived error transmission kernels and error maps to train Convolutional Neural Networks (CNNs). The paper demonstrates the effectiveness of EKDAA by applying it to Fashion MNIST, CIFAR-10, and SVHN benchmarks, showing higher performance than existing biologically plausible algorithms.

**Audience:**

Yes

**Broader Impact Concerns:**

I don't have any concerns.

**Claims And Evidence:**

No

**Requested Changes:**

- Certain sections of the paper, including the introduction, related work, and main methodology, require significant rewriting for enhanced clarity. This should also include fixing formatting issues (e.g., inconsistencies in the use of \citet and \citep) and typos (e.g., in Figure 2, where "(Left:)" should be changed to "(Left)").
- As previously mentioned, I urge the authors to provide more robust justifications for the functionality of their proposed methods. These could take the form of additional theoretical analysis, examples, or comparisons with standard approaches.
- The reported performance of networks learned with backpropagation is surprising, given that they typically approach near 100% accuracy on the training set. Is this a result of the authors using a smaller architecture than is standard for the task, thereby leading to an underparameterized model? Also, the higher performance of EKDAA over networks learned with backpropagation on the FashionMNIST dataset is intriguing and somewhat challenging to accept without additional evidence or explanation.
- It appears that the search ranges for EKDAA and the baseline methods (e.g., Random FA and Direct FA) differ. Could the authors clarify the rationale behind employing different schemes for these approaches? In addition, it would be informative to know how many search spaces were explored for each baseline method.
- It would also be useful to show the learning curves (e.g., number of iterations vs. training loss) to understand the convergence.

**Strengths And Weaknesses:**

The subject matter of this paper aligns with the interests of TMLR's audience, and the authors clearly articulate the work's motivation. However, several weaknesses detract from its overall impact.
- The paper has several vague and not concise statements, making it challenging to follow. There are also numerous formatting issues (for example, inconsistency in the use of \citep and \citet) that disrupt the reading flow and should be addressed.
- The authors should provide a stronger justification for why their proposed method works. The current version of the manuscript lacks a clear motivation for the equations introduced, and there is no justification or proof to support the method's effectiveness.
- I have several concerns regarding the experimental results. These are elaborated in the "Requested Changes" section below.

---

> ### Author Response · Authors · 2023-09-15
> **Response for Reviewer B7BR**
>
> Reviewer B7BR, thank you for your insightful feedback. We have made a number of changes to the manuscript specifically to address clarity and conciseness to the proposed methodology and motivations behind it.
>
> 1. The manuscript has been reviewed for clarity and conciseness to help make it easier to follow. Some of the motivation and claims have been edited for clarity. The formatting issues have also been addressed, fixing how graphics are rendered as well as formatting issues in the text including citation fixes. The manuscript now uses \citep for all references for better reading flow.
>
> 2. The motivation for the work, introduction, related work, and methodology have been reviewed and edited for clarity. Graphics have been added for aiding readers with the motivation behind the work and how the convolutional operators work within the proposed Hebbian learning rule.
>
> 3. The formatting issues in the paper have been resolved. The formatting problems as well as syntactic problems have been reviewed and modified specifically pertaining to the figures and graphics.
>
> 4. We have added a theoretical justification for EKDAA in a new sub section at the end of section 3 Base Rule. In this section we talk about EKDAA’s proposed algorithm for only fully connected layers. We hope this section adds rigor and clarity for readers for the proposed algorithm.
>
> 5. As EKDAA’s novelty is to bring local weight transport to convolutional learning, our models have been designed to focus on determining how learned filters can extract spatial features and use them for classification. To do this, we develop minimal networks with small fully-connected layers so that the networks have to heavily rely on convolutional extracted features rather than focusing an architecture to give optimal accuracy. We have introduced additional figures for FashionMNIST results that showcase EKDAA’s ability to learn convolutional features over time by plotting t-SNE features of the last convolutional layer’s features at random initialization, the final epoch, and every 10 epochs in between shown in Figure 3. Additionally, in Figure 4 we show model convergence for BP and EKDAA on FashionMNIST.
>
> 6. Some additional experiments were run to fill the search range for EKDAA to line up with all other methods, FA/DFA/etc, but did not affect the optimal parameterization. Because Hebbian based algorithms are significantly less studied than backprop, it is still relatively unknown how to parameterize these models so we tested different schemes with a wide range of values that were based on how other bio-inspired papers parameterized their models. As backprop alternatives become researched and used more, there will likely be more common practices and better intuition similar to how backprop is now.  The search spaces for each of the methods analyzed are in the Appendix section D.
>
> 7. The learning curves for FashionMNIST are now in the manuscript in Figure 4. They show the train and test performance with epochs vs. loss. We find that EKDAA has a similar convergence curve to BP even without having clear precisely-calculated gradients.

---

### Review · Reviewer_a4sM · 2023-07-26

**Summary Of Contributions:**

The reviewed work proposes an alternative for the backpropagation with the goal of being more biologically plausible.

The proposed approach, "error-kernel driven activation alignment (EKDAA)" learns "error-kernels" that are used to compute gradient-like quantities to update the network's weights.

The method is shown to perform slightly better than other backdrop alternatives on FMNIST, SVHN, and CIFAR-10.

**Audience:**

Yes

**Broader Impact Concerns:**

As discussed under weaknesses, some of the claims in the broader impacts appear overstated and/or excessively vague.

**Claims And Evidence:**

No

**Requested Changes:**

The concerns discussed as weaknesses above concern the core contribution of the paper. I believe they need to be clarified before the paper is published. I would furthermore encourage the authors to add a section providing a clear discussion of what, mathematically, their criteria for biological plausibility are.

I look forward to the authors' response and hope it will help me better understand their contribution.

**Strengths And Weaknesses:**

The main strength of the paper lies in its empirical improvements over other backprop-free methods. Finding practical Hebbian-like learning rules is an important task for identifying biologically plausible learning rules.

A weakness is that the rational behind the "error-kernels" is poorly explained. The authors write that the error-kernels are " a type of neuron specialized for computing mismatch signals inspired by predictive processing brain theory Clark (2015)," but given that this appears to be the paper's key contribution, a more thorough explanation of the motivation seems in order.

In light of the above, I may misunderstand the method, but I am not sure it lives up to the claims of being Hebbian-like or biologically plausible. My understanding is that a key feature of the Hebbian learning rule is its dependence only on local information. On the other hand, the proposed approach seems to partly mirror backpropagation (see eqns 3-6) by tracing back the signal from the last to the first layer, although using propagation via the error kernels instead of the weight matrices. Thus, the authors' claim that their method "offers a direct, local Hebbian-like, non-differentiable rule, (thus circumventing backprop’s need for a global feedback pathway)" seems misleading.

The claim that their method "works well with systems based on convolution" also seems exagerated since its use loses most of the benefits of convolutional layers in the CIFAR-10 experiments.

Related to the above, there are multiple vague claims of superiority of the proposed method, such as

"Oftentimes, appeasing backprop requires certain constraints that prevent us from easily designing mechanisms that facilitate sparsity, interpretability, adversarial-resistance, probabilistic interpretation, etc. Back-propagation requires computing gradients layer-by-layer enforcing strict requirements for propagation flow such as needing to propagate
in only a feed-forward flow (i.e. lateral connections make no sense for back-propagation). Local
learning mechanisms do not have these restrictions and allow for exploration into novel propagation
flow."

These claims should be substantiated further.
It appears that the proposed method also involves a backward flow of information. Further, reverse mode automatic differentiation can compute the gradient of any function that is the concatenation of differentiable functions so I am not sure what to make of these claims.
Similarly, the authors mention the vanishing/exploding gradient problem but it is unclear if the proposed method circumvents this problem or whether it simply doesn't occur due to the small size of the networks used.

The limited size of the experiments is further explained with "The main limitation of the proposed EKDAA algorithm is currently related its scalability to massive datasets, especially when compared with highly optimized tensor computations
that are implemented in standard deep learning libraries that support backprop-based convolution/deconvolution operations."

I think that this point should be explained in more detail. In particular, since the authors later suggest that the main bottleneck is memory. It is unclear to me what effect optimization has on the

---

> ### Author Response · Authors · 2023-09-15
> **Response for Reviewer A4sM**
>
> Reviewer a4sM, thank you for all of your feedback. We have added in better explanations and have cleaned up much work to increase readability and comprehension.
>
> 1. The rationale behind error kernels has been explained further in 1.1 Bio-plausible Machine Learning.
>
> 2. The intro, related work, and methodology section has been re-written. EKDAA abides by the idea of Hebbian learning by generating local targets for each layer (similar to how the target propagation family of algorithms works) and uses the subtraction of a layer’s target and post-activation to create a local error signal (subtractive Hebbian learning). Because there is no need to calculate a daisy-chained derivative from the output back to the input of the model, we can explore non-differentiable activators. We believe EKDAA is a Hebbian algorithm with the ability to work on non-differentiable functions and does not require a global feedback pathway. EKDAA’s local feedback pathway is in Figure 1.
>
> 3. FashionMNIST to CIFAR-10 involves the move from grayscale to color where there interaction between the red, green and blue channel is necessary to understand in order to learn features. With BP, there are precisely calculated gradients making filter updates easier to optimize to the loss, even when many feature maps are being used to calculate an update. With EKDAA giving approximate error signals, we believe the interaction between channels are harder to optimize on as each individual feature map is getting an approximate error that gets conglomerated when computing deconvolution. Future research on better gradient filter updates is of interest.
>
> 4. All bio-inspired methods we are aware of including TP, FA, Kollen-Pollack, etc. have a backwards flow of information. They focus on creating localized updates rather than using a symmetric global feedback mechanism. BP must adhere to strict rules for differentiation to get an error signal, limiting model design. Local learning methods do not enforce such restrictions and may allow for focusing on the ability to build function-specific models. This has been explained in Broader Impacts.
>
> 5. Auto diff can concatenate a set of differentiable functions to get around dealing with an overall non-differentiable function. However, the idea of concatenating multiple differentiable functions for specific bounds within an activator is not bio-plausible. Activators like signum where either a signal is propagated or killed is bio-plausible in that it behaves like an action potential, where if a threshold is meant an action potential occurs otherwise the signal from an input is terminated (Dale’s law). With EKDAA, the mechanism only requires the pre-activation signal to compute the backwards update, ignoring the activation function completely for computing the error signal, whereas BP must use the post-activation signal and take the differential of it. As EKDAA works with any activator, we believe it allows for the exploration of bio-plausible activators, and we have shown its efficacy to train on color images with a type of bio-plausible activation function. This has been added in section 1.1.
>
> 6. As EKDAA is a subtractive Hebbian rule, the error signals generated do not have the types of drastic weight changes like those seen in unnormalized BP models. However, we don’t currently have clear theoretical proof and our models are still too small to go deep enough to prove it empirically, so we have opted to remove this statement, and will investigate in the future.
>
> 7. EKDAA does require more memory than BP as there are additional matrices required such as the error kernels, but they scale linearly. The current bottleneck in EKDAA is that it is implemented from scratch in Numpy and raw TF tensors. While the current implementation does utilize the GPU, it is significantly slower and less optimized than a TF or PyTorch model would be. This leads to way larger consumptions in memory and in training time due the the custom-built nature to allow EKDAA to run. One key optimization moving forward is to make the convolution and deconvolution operators quicker as they currently use more memory and take longer to compute than TF models. As most of the model is convolutional, this accumulates to significant slow downs over TF models. This work has gone through several rounds of optimization. We started by implementing convolution with the sliding window approach, then moving to a topelitz matrix which had a significant speed up but limited our filter and layer capacity. We moved to FFTs for a speed up and reduction in memory, and moved to using TF's out of the box convolution (just the API function). This is still significantly slower than a TF model. We hope we can put effort in towards better optimization to expand our EKDAA models.
>
> 8. The introduction has been re-written to add bio-plausible principles and criteria for biological plausibility with a graphic in the introduction that will aid readers on our contributions.

---

### Review · Reviewer_BQsB · 2023-08-31

**Summary Of Contributions:**

The authors propose an algorithm for training convolutional neural networks called error-kernel driven activation alignment (EKDAA). EKDAA is motivated from the perspective of biologically plausible credit assignment for CNNs: instead of back propagation, parameter updates are computed using target feature maps calculated from errors that are propagated backwards through the network via error kernels. EKAA is compared to a number of (more than backprop) biologically plausible credit assignment algorithms on FMNIST, CIFAR-10, SVHN and is found to perform slightly better than these alternatives for the architectures used in this paper.  As far as I am aware this is a novel contribution to biologically plausible credit assignment literature.

**Audience:**

Yes

**Claims And Evidence:**

No

**Requested Changes:**

Addressing the major weaknesses listed above would be required for my recommendation for acceptance. Addressing the minor would strengthen the work.

**Strengths And Weaknesses:**

**Strengths**
- EKDAA is a novel credit assignment algorithm and is of interest to the community.
- EKDAA is empirically compared to a number of relevant biologically-inspired learning rules and performs comparably (indeed better for the specific architectures tested).
- EKDAA is tested with a non-differentiable activation function and found to perform well.
- The detailed algorithm is clearly presented.

**Major Weaknesses**
1. It is unclear which components of the proposed algorithm relate specifically to CNNs and which are contributions relevant for biologically plausible credit assignment in general. In my opinion, if these two were disentangled, for example in a section describing and applying EKDAA to fully connected layers, the paper would be strengthened considerably.
2. The paper would be strengthened by a more careful and detailed contextualization of EKDAA with regard to existing work. To my understanding EKDAA is a hybridization of feedback alignment and target propagation: error maps are propagated backwards via random weights (eq 4) and then used to generate targets (eq 6). The authors should more clearly describe how EKDAA relates to existing work.
3. Related to the above points, aside from the empirical results, there are no theoretical justifications given for why EKDAA should result in loss function minimization, nor why the error kernel parameters are updated as in (eq 8).
4. I am concerned as to how strong the baselines are, eg. it is possible to obtain >90% test accuracy for cifar 10 with small resnets.
5. The network architecture is custom to this work and it is unknown how EKDAA will work with other commonly used architectures. The authors rightly point that larger networks will require a larger computational budget, however for general interest I think it is important to demonstrate EKDAA generally performs well with other CNN architectures (for example including also more biologically inspired ones such as cornet). This is essential if the authors wish to claim that EKDAA outperforms other bio-inspired alternatives as they do in the conclusion.
6. The authors state that in EKDAA the forward and backward activities share as minimal information as possible, and as a result this helps the model better overcome poor initialisation and stabilization issues. Could the authors clarify what they mean by information here, and either provide experiments or theory justifying this claim (there are currently neither in my opinion), or remove the statement.

**Minor weaknesses**
- Given that biological plausibility is a key motivating factor a discussion on how deconvolution operation relates to biology would strengthen the paper.
- I expect that EKDAA will not scale to harder datasets like imagenet given the difference between BP on FMNIST vs SVHN.
- Line 4 citation formatting error (Sudhof and Malenka 2008)
- In section 2 related work 2nd paragraph, the authors claim that they are extending hebbian learning to CNNs. I do not think EKDAA approach fits the neuroscience definition of hebbian learning. Could this be clarified?
- In section 2 related work 3rd paragraph the authors list previous work and write that they belong to a different class of algorithms than their work because they require weight transport. This is inaccurate, eg Akrout 2019.
- Statement that CNNs continue to set benchmark standards should be caveated given ViTs are SOTA in many applications.
- Surrogate gradients are an alternative way to “appease backprop-centric optimization” with non-differentiable activation functions. As does Target prop. At times the paper suggests that EKDAA a (novel) solution by omitting this.
- Inference dynamics paragraph, typo “the, the”.
- End of section 2, query typo, should be “EKDAA introduces the notion of error an *map*”.
- No empirical justification or citation is given for the claim that fc layers are not able to learn fashion mnist. Fc layers ((infinite width) are universal function estimators.
- No link to the code.

---

> ### Author Response · Authors · 2023-09-15
> **Response to Reviewer BQsB (Major)**
>
> Reviewer BQsB, thank you for your thorough review. We have incorporated your feedback into the paper. We believe the added suggestions have helped to strengthen the work and present better.
>
> Major 1. In section 3 a new subsection Base Rule has been added in the paper to discuss the proposed algorithm for credit assignment of only connected layers. We hope this will make the proposed algorithm more clear for readers and help justify its learning mechanics.
>
> Major 2. A discussion of how EKDAA relates to other bio-plausible mechanisms like feedback alignment and target propagation has been added in the Representation Alignment section in related work. Feedback alignment and target propagation are compared and contrasted to show how EKDAA relates to existing work.
>
> Major 3. We have added theoretical justification for EKDAA in the new section 3 section Base Rule. This section explains why EKDAA should result in a loss function minimization during training.
>
> Major 4. Our performance results for BP and EKDAA are computed with a custom built library to provide an apples-to-apples comparison since we implemented  EKDAA from scratch; it would not be a fair comparison to compare with a Tensorflow model for BP since Tensorflow has numerous behind-the-scenes optimizations and enhancements that make neural networks stable and easy to train. To make a fair comparison, we compare BP and EKDAA within our handwritten library. We also focus on creating models consisting mostly of convolutional features with few small fully-connected layers so that the model performance relies more heavily on the filter updates.
>
> Major 5. This paper introduces the EKDAA learning mechanism and focuses on a feed-forward classification network only. The model architecture was selected to have multiple layers of convolution (as deep as the current implementation could handle with the amount of GPU resources we have), while restricting the fully-connected layers to be quite small. With enhanced optimization of our implementation, we wish to expand EKDAA to other network types including a residual network. Introducing skip connections into the model would be an easy modification as it just repeats the signal from a previous layer to a future layer, and does not require modification to the error kernel computation. However, in the original He et al. paper, they focus on very deep networks (18 layers to 152 layers). Our current deepest model is only 6 layers of convolution and the model is unlikely to see any gain from the reintroduction of a signal as it is likely not deep enough to have the signal degradation to be mitigated. Other bio-inspired structures such as the one introduced with CORnet would be a step forwards towards having a bio-inspired neural structure with a bio-inspired learning mechanism for training. CORnet’s inspiration to model the visual cortical areas within the mammalian brain would bring the area of bio-inspired learning mechanisms closer towards more brain-like modeling ability. With the invention of unrolling in time for EKDAA, a CORnet trained with EKDAA rather than BP should be possible. The conclusion of the paper has been altered to specify that EKDAA outperforms the bio-inspired alternatives in the networks tested for the task of image classification. We also have added a discussion into other imaging tasks that we want to explore in future work.
>
> Major 6. With EKDAA, the backward signal creation does not use the post-activation gradients from the forward pass (as it is not computing gradients from them). It instead creates its own layerwise targets and uses error kernel projections to transfer error signals upwards through a model for updates. With BP, if there was a poor weight initialization schema, for example all initial weights were assigned to zero, then the model will be unable to train since the gradients will be killed when differentiating by zeros from the post-activation incoming neurons. However, by generating local layerwise targets with EKDAA, even if the incoming pre-activation are all zeros, it will still give an error signal to learn from as the error signal is the difference in target versus actual (subtractive Hebbian learning) rather than differentiable. This inherent property of EKDAA will allow for model training even with poor initialization schemas. This reasoning has been put into the paper and that section has been re-written.

---

> ### Author Response · Authors · 2023-09-15
> **Response to Reviewer BQsB (Minor)**
>
> Minor 1. A discussion on how the convolution and deconvolution operators adhere to bio-plausibility was an oversight. We have added this into the paper to make convolution and deconvolution rational choices towards creating bio-plausible vision models.
>
> Minor 2. We aren’t sure how EKDAA will scale yet. Other biologically-plausible algorithms have generally not scaled to large datasets that well yet, but it is an essential task to work on as the BP alternative space grows. However, we have trained on natural color images in this work, giving us a starting point to focus on scaling in future work by modifying how the mechanism works, understanding the parameter space better, and being able to optimize the implementation itself for reduced computational requirements.
>
> Minor 3. The citation error in Line 4 has been resolved and all of the other citations have been reviewed for correctness.
>
> Minor 4. This extension of Hebbian learning to convolutional models was not explained clearly in the paper before. The introduction has been re-written to introduce what kinds of bio-plausible principles can be brought into neural networks in general with a specific emphasis on our contribution in the bio-plausible space. We specifically create a Hebbian-based algorithm that mitigates the weight transport problem (i.e. the idea that using the feed-forward weights to calculate the error signal is not biologically plausible) and works for imaging data with a convolutional approach. Our proposed approach, EDKAA, allows for working with more bio-plausible activation functions like the signum function which abides by Dale’s law.
>
> Minor 5. In Akrout 2019, their work did not require weight transport. This was an error that has been resolved in the manuscript now.
>
> Minor 6. As you mentioned, Visual Transformers (ViTs) have shown significant effectiveness in recent years with state-of-the-art performance on imaging tasks. The statement regarding CNNs setting benchmark standards has been altered to include this in conjunction with CNNs about standard methodologies for solving computer vision problems.
>
> Minor 7. The end of the introduction section that suggests EKDAA is novel in appeasing backprop-centric optimization has been modified to not say exclusively that this is a property of EKDAA but of all Hebbian based and non-derivative algorithms.
>
> Minor 8 & 9. The “the the” type has been fixed in the inference dynamics section. The other typo regarding “EKDAA introduces the notion of error an map” at the end of the second section has also been resolved. The manuscript has been reviewed and edited to fix all other found typos.
>
> Minor 10. We trained FashionMNIST for backpropagation and EKDAA for both solely fully connected as well as convolutional models, and we see that convolution benefits for both types of algorithms. While fully connected layers are universal function approximators, we could not empirically get any fully connected models for backpropagation or EKDAA to train any better than the reported results even with wider or more layers. The extensive analysis of backpropagation has also shown as image datasets become harder feature extraction methods like convolution or transformers become more and more necessary to learn quality discriminating features for classification.
>
> Minor 11. A link to our codebase is now in the manuscript in the second to last paragraph of the technical implementation. The link to the codebase can also be found  at https://anonymous.4open.science/r/EKDAA_release-5613/. We have used the anonymous version of github to remain double blind, and will switch back to the regular github link upon publication.

---

### Decision · Action_Editors · 2023-10-09

**Recommendation:** Accept with minor revision

**Comment:**

The reviewers highlighted a number of concerns around clarity, justification, and some of the evaluations. The authors did a sufficiently good job of attending to these concerns to lead all three reviewers to recommend acceptance. However, the authors should perform a final check to make sure that they have incorporated all of the outstanding changes promised (e.g. switching to a regular GitHub codebase) before final acceptance.

**Audience:**

Yes, those TMLR readers interested in local learning rules and neuroscience would find this article of interest.

**Claims And Evidence:**

In this work the authors introduce a learning algorithm for more biologically realistic learning in deep neural networks that avoids the "weight transport problem" by learning kernels to propagate local error signals, which they call error-kernel driven activation alignment (EKDAA). The authors claim that the algorithm is effective at learning in deep convolutional networks while not relying on weight transport. As evidence, they show that it achieves roughly the same performance as backpropagation, and better than other bio-plausible algorithms, on the Fashion MNIST, CIFAR-10 and SVHN benchmarks, and that it can extract relevant visual features. This evidence was reasonably convincing and fairly clear.

---

> ### Author Response · Authors · 2023-11-06
> **Thank you and final revisions**
>
> We would like to again thank our reviewers for their valuable feedback on our work. The suggestions provided by all reviewers helped us to refine and clean up the paper, as well as strengthen it. Please let us know if we need to do anything further.
>
> We have uploaded the final camera-ready version of our paper, along with the link to our codebase, and a link to the video presentation.